# The impact of skeletal muscle disuse on distinct echo intensity bands: A retrospective analysis

Zachary S. Logeson[1], Rob J. MacLennan[2], Gerard-Kyle B. Abad[1], Johnathon M. Methven[1], Molly R. Gradl[1], Matheus D. Pinto[3], Ronei S. Pinto[4], Matt S. Stock[1]*

1 Neuromuscular Plasticity Laboratory, Institute of Exercise Physiology and Rehabilitation Science, University of Central Florida, Orlando, Florida, United States of America, 2 Applied Neuromuscular Physiology Laboratory, Oklahoma State University, Stillwater, Oklahoma, United States of America, 3 Centre for Exercise and Sport Science Research (CESSR), School of Medical and Health Sciences, Edith Cowan University, Joondalup, Australia, 4 Exercise Research Laboratory, Physical Education, Physiotherapy and Dance School, Universidade Federal do Rio Grande do Sul, Porto Alegre, Brazil

* matt.stock@ucf.edu

**Data Availability Statement:** All relevant data are within the paper and its Supporting Information files.

## Abstract

Echo intensity (EI) is a novel tool for assessing muscle quality. EI has traditionally been reported as the mean of the pixel histogram, with 0 and 255 arbitrary units (A.U.) representing excellent and poor muscle quality, respectively. Recent work conducted in youth and younger and older adults suggested that analyzing specific EI bands, rather than the mean, may provide unique insights into the effectiveness of exercise and rehabilitation interventions. As our previous work showed deterioration of muscle quality after knee joint immobilization, we sought to investigate whether the increase in EI following disuse was limited to specific EI bands. Thirteen females (age = 21 yrs) underwent two weeks of left knee immobilization and ambulated via crutches. B-mode ultrasonography was utilized to obtain images of the immobilized vastus lateralis. The percentage of the total number of pixels within bands of 0–50, 51–100, 101–150, 151–200, and 201–255 A.U. was examined before and after immobilization. We also sought to determine if further subdividing the histogram into 25 A.U. bands (i.e., 0–25, 26–50, etc.) would be a more sensitive methodological approach. Immobilization resulted in a decrease in the percentage of pixels within the 0–50 A.U. band (-3.11 ± 3.98%), but an increase in the 101–150 A.U. (2.94 ± 2.64%) and 151–200 A.U. (0.93 ± 1.42%) bands. Analyses of variance on the change scores indicated that these differences were large and significant ($\%EI_{0-50}$ vs. $\%EI_{101-150}$: $p < .001$, $d = 1.243$); $\%EI_{0-50}$ vs. $\%EI_{151-200}$: $p = .043$, $d = 0.831$). The effect size for the $\%EI_{51-100}$ versus $\%EI_{101-150}$ comparison was medium/large ($d = 0.762$), but not statistically significant ($p = .085$). Further analysis of the 25 A.U. bands indicated that the percentage of pixels within the 25–50 A.U. band decreased (-2.97 ± 3.64%), whereas the 101–125 (1.62 ± 1.47%) and 126–150 A.U. (1.18 ± 1.07%) bands increased. Comparison of the 50 A.U. and 25 A.U. band methods found that 25 A.U. bands offer little additional insight. Though studies are needed to ascertain the factors that may influence specific bands, changes in EI during muscle disuse are not homogeneous across the pixel histogram. We encourage investigators to think critically about the

**Funding:** Funding support for this project was provided to RJM by the De Luca Foundation and the University of Central Florida's Office of Research to MSS. Article processing charges were provided to ZSL in part by the UCF College of Graduate Studies Open Access Publishing Fund. The funders had no role in study design, data collection and analysis, decision to publish, or preparation of the manuscript.

**Competing interests:** The authors have declared that no competing interests exist.

robustness of data obtained from EI histograms, rather than simply reporting the $EI_{mean}$ value, in muscle quality research.

## Introduction

Skeletal muscle atrophy is an inevitable consequence of prolonged disuse during detraining, injury, and a variety of disease states [1,2]. Disuse does not merely affect muscle morphology but is also associated with muscle weakness and functional impairments [3,4]. Populations of concern that are prone to the deleterious effects of muscle disuse include, but are not limited to, patients undergoing surgical procedures, immobilization, or prolonged bed rest. Concerns about disuse are magnified in older adults because muscle weakness is a strong, independent predictor of mortality [5]. In athletic populations, females tend to be more susceptible to the effects of joint immobilization compared to males, especially at the knee extensor muscle group [6–10].

Although the effects of disuse on muscle size, strength, and functional capacity have been well established, changes in echo intensity (EI), which is used as a surrogate of skeletal muscle quality, following immobilization has recently garnered attention from the scientific community [8,11]. Magnetic resonance imaging (MRI), computed tomography (CT), and B-mode ultrasonography have all been used to assess EI in research. Among these techniques, ultrasonography has the highest benefit-cost ratio, and the information it provides is similar to that of MRI [12,13]. In addition to measures of muscle size, EI can be assessed quickly from ultrasound scans [14]. To do so, images are taken of a muscle of interest and are evaluated by quantifying the degree of brightness/darkness of a region on the image. Using ImageJ software, EI exists on a scale of 0–255 arbitrary units (A.U.) in which skeletal muscle tissue appears black ("hypoechoic", EI = 0) and non-contractile tissues (i.e., intramuscular adipose and fibrous tissue) appears white ("hyperechoic", EI = 255). Although the exact basis for EI has yet to be formally verified, previous investigations have demonstrated a strong correlation between EI and fibrous tissue content [15,16], as well as intramuscular adipose tissue [12,14].

Despite widespread interest in EI, there are a variety of gaps in knowledge concerning optimal approaches during image analysis [17]. For example, EI has been traditionally calculated as the mean grayscale histogram value ($EI_{mean}$) on the 0–255 A.U. scale. However, analysis of a single measure of central tendency from a spectrum of values might provide limited information. Recently, Pinto and Pinto [18] presented an alternative method of assessing the concentrations of pixels within different clusters of the grayscale histogram by examining EI intervals of 50 A.U. bands (e.g., 0–50, 51–100, 101–150, 151–200, 201–255 A.U.). Pinto and Pinto [18] reported that rectus femoris $EI_{mean}$ values increased with age when using the traditional method of $EI_{mean}$ to assess muscle quality between adolescents and younger and older adults (mean ages of 13, 32, and 67 years, respectively). However, when using the EI bands method, they were able to identify higher values in EI bands between the ranges of 0–50 in youth when compared to younger and older adults [18]. EI values were also positively correlated with age in bands ≥ 51–100 [18]. The different conclusions that were drawn between the two methodological approaches illustrated that the EI bands method may provide more insight on the specific tissue changes in each of the populations of interest. As such, Pinto and Pinto [18] suggested that the analysis of specific EI bands, rather than simply analyzing the $EI_{mean}$, might help researchers further their understanding of changes in muscle quality during a variety of conditions or interventions.

Work from our laboratory recently demonstrated that two weeks of left knee joint immobilization in young women resulted in declines in vastus lateralis muscle size and quality [8].

Specifically, the increase in $EI_{mean}$ (indicating worsening muscle quality) was greater in magnitude than the decrease in muscle cross-sectional area (EI Cohen's $d$ = 0.918; muscle cross-sectional area $d$ = 0.570). We also reported a strong association between the increase in vastus lateralis $EI_{mean}$ and the decrease in cross-sectional area ($r$ = —.649). These findings led us to conclude that $EI_{mean}$ appears to be an attractive clinical tool for monitoring muscle quality during extended periods of disuse. Given the need for advanced methodological approaches for analyzing muscle quality [17] and the compelling results presented by Pinto and Pinto [18], in this paper, we sought to retrospectively explore whether the increase in EI following two weeks of knee joint immobilization was limited to specific bands of the EI histogram [8]. We hypothesized that the percentage of pixels in bands 0–50 A.U. would decrease, while the percentage of pixels in bands 51–100, 101–150, 151–200, and 201–255 A.U. would increase or remain unchanged. As an exploratory aim, we also sought to determine whether analysis of narrower EI bands (0–25, 26–50, 51–75 A.U., etc.) would provide a more complete and descriptive understanding of changes in muscle quality during disuse.

## Materials and methods

### Experimental approach to the problem

The data presented herein represent a retrospective analysis of vastus lateralis results published previously [8]. For full details, including results from the non-immobilized limb, the rectus femoris muscle, and muscle cross-sectional area measurements, the reader is directed to the publication by MacLennan et al [8]. In brief, this investigation utilized a within-participants design in which college-aged females voluntarily underwent two weeks of left knee joint immobilization and ambulated with crutches. Before (PRE) and following (POST) the two-week immobilization protocol, B-mode ultrasonography images of the vastus lateralis were obtained.

### Participants

Data from thirteen college-aged females (mean ± SD age = 21 ± 2 years, mass = 61.6 ± 4.6 kg, height = 164.7 ± 6.1 cm) were included in our study's final analysis. Major exclusion criteria included (but were not limited to) a history of blood clots, use of oral contraceptives, body mass index > 30 $kg/m^2$, and surgery on the lower limbs within the past year. Recruitment efforts included marketing on laboratory and university websites, social media posts, presentations in group settings and use of flyers. All participants read, understood, and signed an informed consent form prior to involvement in the study. The University of Central Florida's Institutional Review Board approved the study protocol (Study # BIO-17-13642). All methods were carried out in accordance with the guidelines and regulations described by the Declaration of Helsinki. Participants were compensated $350 for completing the study.

### Immobilization procedures

Each participant was properly fitted for the use of a knee joint immobilization brace (T Scope® Premier Post-Op Knee Brace, Breg, Inc., Carlsbad, CA, USA) during the pretesting data collection session. Participants were instructed to always wear the brace, except when sleeping, and to keep the brace locked to prevent weight bearing through the left lower extremity (Fig 1). Participants were also provided a stocking (Rolyan Extra Soft Stockinette, 100% Cotton, Performance Health, Warrenville, IL, USA) to wear underneath the brace at all times in order to prevent skin breakdown and improve comfort. Compression stockings (Medi-Pak Anti-Embolism Stockings, McKesson, San Francisco, CA, USA) were distributed for nighttime

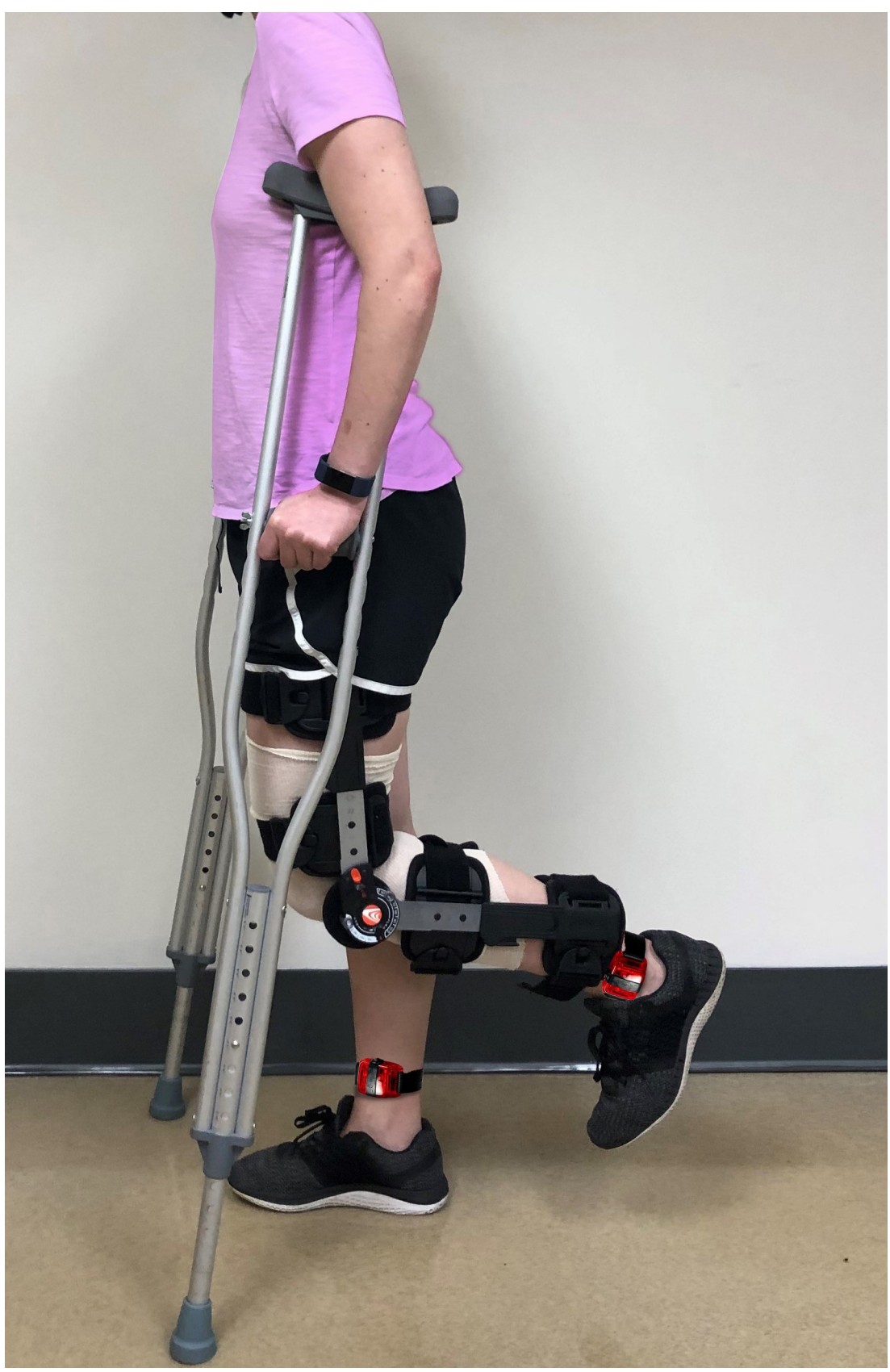

**Fig 1. An example of the equipment and procedures utilized for the knee joint immobilization protocol.**

use to prevent blood clots. Training was provided for each participant on the proper use of axillary crutches while in the community, including stair and curb navigation. Each participant was provided with a handout and access to a YouTube video containing instructions on how to properly perform light range of motion exercises while lying supine in bed for the involved ankle and knee in an effort to prevent muscle contractures and minimize the risk of vascular pathology [6,7]. All participants performed these exercises twice daily (morning and evening). Study investigators text messaged or spoke via telephone to each participant daily to ensure compliance. Compliance with the immobilization procedures was objectively verified through the use of Actigraph GT9X accelerometers (ActiGraph Inc, Penscola, FL, USA), which were worn around both ankles throughout the study.

## Ultrasonography measurements and analysis

Before images were taken, participants remained supine for five minutes to allow fluid redistribution in the target muscle [19]. A portable B-mode imaging device (GE Logiq e BT12, GE Healthcare, Milwaukee, WI, USA) and a multi-frequency linear-array probe (12 L-RS, 5–13 MHz, 38.4-mm field of view; GE Healthcare, Milwaukee, WI, USA) was utilized to capture transverse images with the panoramic function (LogiqView, GE Healthcare, Milwaukee, WI, USA) for the vastus lateralis. In accordance with previous studies, images were captured at 50% of the length between the greater trochanter and the lateral femoral condyle [18]. During femur measurement, markings were placed on the skin at the level to be imaged. A dense foam pad was strapped onto the leg at the site to guide the ultrasound probe. Water-soluble transmission gel (Aquasonic 100 ultrasonography transmission gel, Parker Laboratories, Inc., Fairfield, NJ, USA) was applied to the skin along the probe path to enhance transducer coupling during imaging. Ultrasound settings were kept constant at both time points, including frequency (10 MHz), gain (55dB), and dynamic range (72). Three images of the vastus lateralis were taken during both pretest (PRE) and posttest (POST). All results represent the mean values from the three images.

ImageJ was used to examine each ultrasound image. The polygon function was then utilized to outline the vastus lateralis just under the fascial plane to select as much muscle as possible while avoiding subcutaneous adipose tissue and surrounding fascia [3,18]. The histogram produced from the ImageJ software was exported to a Microsoft Excel spreadsheet in order to express EI bands in two separate intervals: increments of 50 A.U. (0–50, 51–100, 101–150, 150–200, and 201–255 A.U.) and increments of 25 A.U. (0–25, 26–50, 51–75, 76–100, 101–125, 126–150, 151–175, 176–200, 201–225, and 226–255 A.U.). The quantity of pixels in each band was then expressed as a percentage of the total pixel content in the EI histogram. Expressing the number of EI pixels within each band as a percentage of the total number allows for normalization of images possessing varying numbers of pixels. Our laboratory has used these procedures in multiple studies, with excellent test-retest reliability statistics published elsewhere [20]. An example of the ultrasonography image analysis procedures has been displayed in Fig 2.

## Statistical analyses

All statistical analyses were performed on change scores (posttest-pretest) within each band. Two separate one-way repeated measures (0–50, 51–100, 101–150, 150–200, 201–255 A.U.; 0–25, 26–50, 51–75, 76–100, 101–125, 126–150, 151–175, 176–200, 201–225, 226–255 A.U.)

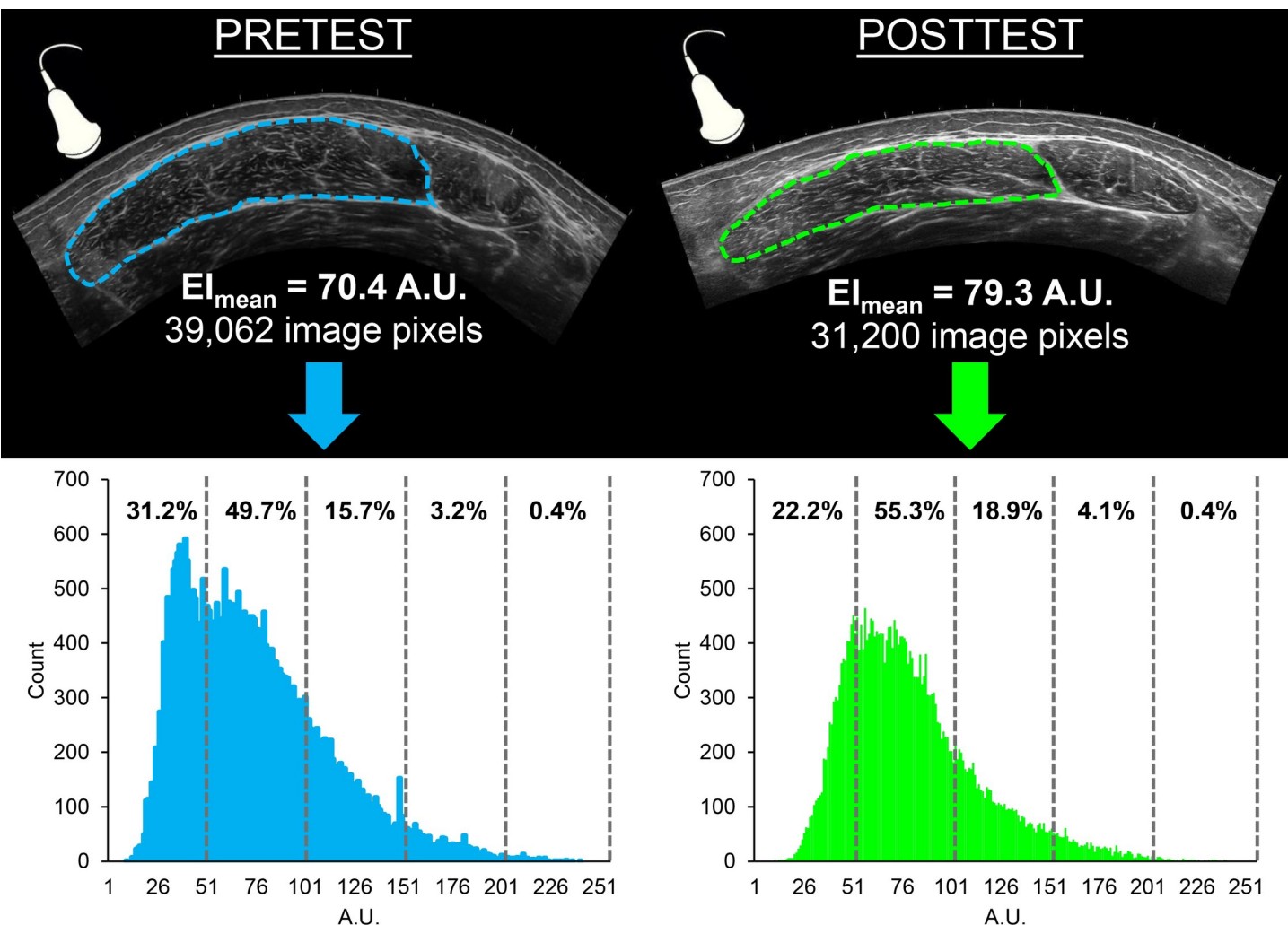

**Fig 2. Example pretest (left) and posttest (right) B-mode ultrasonography images of the immobilized vastus lateralis for one participant.** The borders of the vastus lateralis muscle have been outlined. The corresponding EI histograms, with the percentage of total number of pixels for each region (i.e., 0–50, 51–100, 101–150, 151–200, and 201–255 A.U.) are shown beneath each respective image. Note the shift in the histogram observed following immobilization.

analyses of variance (ANOVAs) were used to examine mean differences in the change scores across bands. If the sphericity assumption was violated, Greenhouse-Geisser corrections were applied. The partial eta squared statistic ($\eta^2$) was used as a measure of the effect size for each repeated measures ANOVA, with values of 0.01, 0.06, and 0.14 representing small, medium, and large effects, respectively [21]. If the ANOVA was significant, follow-up analyses included Bonferroni post-hoc comparisons. We also evaluated 95% confidence intervals for mean differences (CIs) and effect sizes via Cohen's *d* statistics when examining pairwise differences. Small, medium, and large Cohen's *d* corresponded to values of 0.20, 0.50, and 0.80, respectively [22]. An alpha level of 0.05 was used to determine statistical significance for all repeated measures ANOVAs and Bonferroni post hoc comparisons. All statistical procedures were carried out with JASP software (version 0.14.1, University of Amsterdam, Amsterdam, The Netherlands).

**Table 1. Mean (SD) pretest, posttest, and change score values within each of the specified echo intensity bands for the immobilized vastus lateralis.**

| | 50 A.U. Bands | | | | |
|---|---|---|---|---|---|
| | **0–50** | **51–100** | **101–150** | **151–200** | **201–255** |
| Pretest (%) | 16.27 (9.97) | 59.73 (5.98) | 21.51 (6.51) | 3.55 (1.32) | 0.33 (0.21) |
| Posttest (%) | 13.16 (8.29) | 58.96 (5.48) | 24.45 (7.17) | 4.48 (2.04) | 0.44 (0.31) |
| % Change (Posttest-Pretest) | -3.11 (3.98) | -0.77 (4.74) | 2.94 (2.64) | 0.93 (1.42) | 0.12 (0.17) |

| | 25 A.U. Bands | | | | | | | | | |
|---|---|---|---|---|---|---|---|---|---|---|
| | **0–25** | **26–50** | **51–75** | **76–100** | **101–125** | **126–150** | **151–175** | **176–200** | **201–225** | **226–255** |
| Pretest (%) | 0.44 (0.80) | 15.83 (9.31) | 31.38 (4.83) | 28.35 (5.22) | 13.97 (4.33) | 6.15 (2.16) | 2.54 (0.88) | 1.01 (0.47) | 0.27 (0.17) | 0.06 (0.06) |
| Posttest (%) | 0.29 (0.62) | 12.87 (7.87) | 31.15 (7.53) | 27.81 (8.22) | 15.63 (4.48) | 7.33 (2.65) | 3.16 (1.31) | 1.32 (0.75) | 0.37 (0.26) | 0.07 (0.06) |
| % Change (Posttest-Pretest) | -0.14 (0.53) | -2.97 (3.68) | -0.23 (6.55) | -0.54 (5.78) | 1.65 (1.53) | 1.18 (1.12) | 0.62 (0.84) | 0.31 (0.59) | 0.10 (0.17) | 0.02 (0.04) |

All values are expressed as a percentage of the total number of image pixels.

## Results

Table 1 displays the mean ± SD pretest, posttest, and change score values for both the 50 A.U. and 25 A.U. bands.

### 50 A.U. intervals

The Greenhouse-Geisser corrected one-way repeated measures ANOVA was statistically significant ($F = 5.428$, $p = .014$) and the effect size was large ($\eta_p^2 = .311$). The Bonferroni post-hoc comparisons demonstrated that the change in $\%EI_{0-50}$ was significantly greater and larger in magnitude than $\%EI_{51-100}$ ($p < .001$, $d = 1.243$, 95% CI = -10.02 –-2.08 A.U.) and $\%EI_{101-150}$ ($p = .043$, $d = 0.831$, 95% CI = -8.01–0.07 A.U.). The effect size for the $\%EI_{51-100}$ versus $\%EI_{101-150}$ comparison was medium/large ($d = 0.762$), but not statistically significant ($p = .085$, 95% CI = -7.68–0.27 A.U.). Table 2 displays the results from all possible Bonferroni pairwise comparisons.

### 25 A.U. intervals

The Greenhouse-Geisser corrected one-way repeated measures ANOVA was not statistically significant ($F = 1.889$, $p = .176$) and the effect size was medium/large ($\eta_p^2 = .136$). The greatest

**Table 2. 50 A.U. band pairwise comparisons following one-way ANOVA.**

| | | | 95 CI for Mean Difference | | | |
|---|---|---|---|---|---|---|
| | | **Mean Difference** | **Lower** | **Upper** | **Cohen's *d*** | ***p*** |
| 0–50 | 51–100 | -2.34 | -6.31 | 1.63 | -0.481 | 0.892 |
| | 101–150 | -6.05 | -10.02 | -2.08 | -1.243 | < .001* |
| | 151–200 | -4.04 | -8.01 | -0.07 | -0.831 | 0.043* |
| | 201–255 | -3.23 | -7.20 | 0.74 | -0.663 | 0.207 |
| 51–100 | 101–150 | -3.71 | -7.68 | 0.27 | -0.762 | 0.085 |
| | 151–200 | -1.70 | -5.67 | 2.27 | -0.350 | 1.000 |
| | 201–255 | -0.89 | -4.89 | 3.08 | -0.182 | 1.000 |
| 101–150 | 151–200 | 2.01 | -1.97 | 5.9 | 0.412 | 1.000 |
| | 201–255 | 2.82 | -1.152 | 6.790 | 0.579 | 0.421 |
| 151–200 | 201–255 | 0.81 | -3.158 | 4.784 | 0.167 | 1.000 |

* = statistically significant.

change was observed for the %EI$_{26-50}$ band (2.97 A.U.), and all effect sizes for the pairwise differences were medium or large (*d* range = 0.529–1.01). Table 3 displays the results from all possible Bonferroni pairwise comparisons.

## Discussion

Our laboratory recently reported that two weeks of left knee joint immobilization resulted in a large increase in EI$_{mean}$ that was greater in magnitude than the decrease in cross-sectional area of the vastus lateralis, highlighting the potential utility of EI for studying clinically relevant topics [8]. To overcome some of the analytical limitations of the traditional EI$_{mean}$ method, Pinto and Pinto [18] introduced a novel approach involving assessment of distinct EI bands, rather than a single measure of central tendency from the entire histogram. The present study tested the hypothesis that application of the EI band approach to knee joint immobilization data would show distinct patterns of change across vastus lateralis EI bands, rather than a homogenous response. Our findings showed that immobilization resulted in a decrease in the percentage of pixels within the 0–50 A.U. band (-3.11 ± 3.98%), but an increase in the 101–150 A.U. (2.94 ± 2.64%) and 151–200 A.U. (0.93 ± 1.42%) bands. Analysis of the 50 A.U. change scores indicated that these differences were large. Similarly, a medium/large effect size for the %EI$_{51-100}$ versus %EI$_{101-150}$ comparison was noted. Further analysis of the 25 A.U. bands indicated that the percentage of pixels within the 25–50 A.U. band decreased (-2.97 ± 3.64%), whereas the 101–125 (1.62 ± 1.47%) and 126–150 A.U. (1.18 ± 1.07%) bands increased. We also found that the concentration of pixels in the 0–50, 51–100, and 101–150 A.U. bands contained ~95% of pixels from the entire histogram. Collectively, these findings suggest that analysis of distinct EI bands, rather than the traditional EI$_{mean}$ approach, may provide unique insights into changes in muscle quality during disuse.

EI has shown to be strongly associated with both intramuscular adiposity [12,14] and fibrous tissue content [15,16]. As such, EI has become widely adopted in the literature as an estimate of skeletal muscle quality and is used to study aging [23], muscle strength/weakness [24], and body composition [25]. To date, each of these studies have relied on the traditional EI$_{mean}$ approach. To our knowledge, the work by Pinto and Pinto [18] was the first to question whether more information can be obtained by scrutinizing separate regions of the traditional 0–255 A.U. histogram. Their work illustrated unique age group × band interactions, with youth showing a high percentage of pixels within %EI$_{0-50}$ band, but older adults possessing their highest percentage of pixels within the %EI$_{51-100}$ band. These changes did not follow the age-related differences in EI$_{mean}$. While the present study was limited to young females, our finding that the %EI$_{0-50}$ band was particularly influenced by disuse seems to parallel the age-related differences noted by Pinto and Pinto [18]. The reason(s) for the large change in the %EI$_{101-150}$ band, but not the %EI$_{51-100}$ band, is unclear. We speculate that, as fat and/or fibrous tissue entered the muscle, pixels in each band shifted to higher bands. For example, a large number of pixels may have left the %EI$_{0-50}$ band and entered the %EI$_{51-100}$ band. If there is a cascading effect, it is possible that enough pixels left the %EI$_{51-100}$ band to increase the number in the %EI$_{101-150}$ band, but those leaving the %EI$_{51-100}$ band were replaced by pixels from the %EI$_{0-50}$ band. This seems logical; however, it is not known whether the explanation is so simple and it requires further study.

An exploratory aim of this study was to determine if analysis of narrower EI bands (0–25, 26–50, 51–75 A.U., etc.) would provide a more complete understanding of changes in muscle quality during disuse. Given the results of the 50 A.U. increment analysis, one might expect that further subdividing the EI bands would assist in determining the most important region of the EI histogram for studying disuse. Surprisingly, PRE-POST differences with the narrower

**Table 3. 25 A.U. band pairwise comparisons following one-way ANOVA.**

| | | Mean Difference | Lower | Upper | Cohen's d | p |
|---|---|---|---|---|---|---|
| 0–25 | 26–50 | 2.82 | -1.12 | 6.76 | 0.640 | 0.820 |
| | 51–75 | 0.09 | -3.85 | 4.03 | 0.020 | 1.000 |
| | 76–100 | 0.39 | -3.55 | 4.34 | 0.089 | 1.000 |
| | 101–125 | -1.80 | -5.74 | 2.15 | -0.407 | 1.000 |
| | 126–150 | -1.33 | -5.27 | 2.62 | -0.301 | 1.000 |
| | 151–175 | -0.76 | -4.70 | 3.18 | -0.173 | 1.000 |
| | 176–200 | -0.46 | -4.40 | 3.48 | -0.104 | 1.000 |
| | 201–225 | -0.25 | -4.19 | 3.69 | -0.056 | 1.000 |
| | 226–255 | -0.16 | -4.10 | 3.78 | -0.036 | 1.000 |
| 26–50 | 51–75 | -2.74 | -6.68 | 1.21 | -0.620 | 0.991 |
| | 76–100 | -2.43 | -6.37 | 1.51 | -0.551 | 1.000 |
| | 101–125 | -4.62 | -8.56 | -0.68 | -1.047 | 0.007* |
| | 126–150 | -4.15 | -8.09 | -0.21 | -0.941 | 0.028* |
| | 151–175 | -3.59 | -7.53 | 0.36 | -0.813 | 0.131 |
| | 176–200 | -3.28 | -7.22 | 0.66 | -0.744 | 0.283 |
| | 201–225 | -3.07 | -7.01 | 0.87 | -0.696 | 0.468 |
| | 226–255 | -2.98 | -6.93 | 0.96 | -0.676 | 0.572 |
| 51–75 | 76–100 | 0.31 | -3.63 | 4.25 | 0.070 | 1.000 |
| | 101–125 | -1.88 | -5.83 | 2.06 | -0.427 | 1.000 |
| | 126–150 | -1.41 | -5.35 | 2.53 | -0.320 | 1.000 |
| | 151–175 | -0.85 | -4.79 | 3.09 | -0.193 | 1.000 |
| | 176–200 | -0.54 | -4.49 | 3.40 | -0.123 | 1.000 |
| | 201–225 | -0.33 | -4.28 | 3.61 | -0.076 | 1.000 |
| | 226–255 | -0.25 | -4.19 | 3.69 | -0.056 | 1.000 |
| 76–100 | 101–125 | -2.19 | -6.13 | 1.75 | -0.497 | 1.000 |
| | 126–150 | -1.72 | -5.66 | 2.22 | -0.390 | 1.000 |
| | 151–175 | -1.16 | -5.10 | 2.79 | -0.262 | 1.000 |
| | 176–200 | -0.85 | -4.79 | 3.09 | -0.193 | 1.000 |
| | 201–225 | -0.64 | -4.58 | 3.30 | -0.145 | 1.000 |
| | 226–255 | -0.56 | -4.50 | 3.39 | -0.126 | 1.000 |
| 101–125 | 126–150 | 0.47 | -3.47 | 4.41 | 0.107 | 1.000 |
| | 151–175 | 1.03 | -2.91 | 4.98 | 0.234 | 1.000 |
| | 176–200 | 1.34 | -2.60 | 5.28 | 0.304 | 1.000 |
| | 201–225 | 1.55 | -2.39 | 5.49 | 0.351 | 1.000 |
| | 226–255 | 1.64 | -2.31 | 5.58 | 0.371 | 1.000 |
| 126–150 | 151–175 | 0.56 | -3.38 | 4.51 | 0.128 | 1.000 |
| | 176–200 | 0.87 | -3.07 | 4.81 | 0.197 | 1.000 |
| | 201–225 | 1.08 | -2.86 | 5.02 | 0.245 | 1.000 |
| | 226–255 | 1.17 | -2.78 | 5.11 | 0.264 | 1.000 |
| 151–175 | 176–200 | 0.31 | -3.64 | 4.25 | 0.069 | 1.000 |
| | 201–225 | 0.52 | -3.43 | 4.46 | 0.117 | 1.000 |
| | 226–255 | 0.60 | -3.34 | 4.54 | 0.136 | 1.000 |
| 176–200 | 201–225 | 0.21 | -3.73 | 4.15 | 0.048 | 1.000 |
| | 226–255 | 0.30 | -3.65 | 4.24 | 0.067 | 1.000 |
| 201–225 | 226–255 | 0.09 | -3.86 | 4.03 | 0.020 | 1.000 |

* = statistically significant.

## Band-Specific Changes in Echo Intensity

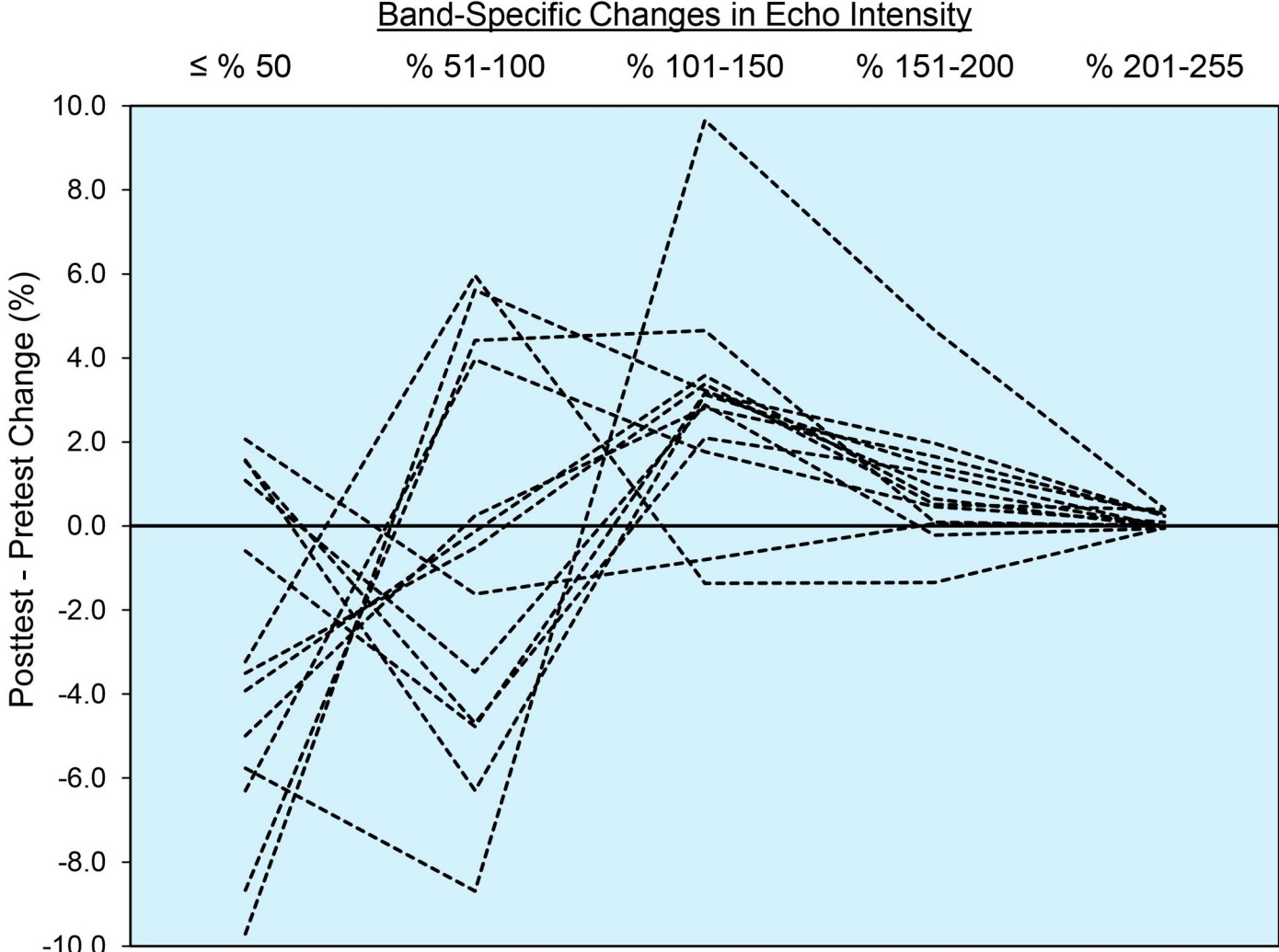

**Fig 3. Individual participant change score values across each of the 50 A.U. bands.**

bands were smaller and the repeated measures ANOVA on the change scores was not statistically significant. We suspect that these discrepant findings could be related to the fact that splitting the 50 A.U. bands into two separate 25 A.U. bands may have introduced additional variability, thereby decreasing statistical power. For example, close analysis of the $\%EI_{0-25}$ band showed that some images have 0 pixels in this region, whereas others had $\geq 2\%$. As this aim was exploratory, it is important to acknowledge that our decision to analyze EI bands in intervals of 25 A.U. was somewhat arbitrary and meant to reflect the values utilized by Pinto and Pinto [18]. Whether there is an optimal band to study in muscle quality research, rather than an arbitrary range, is a question that deserves further consideration from investigators. Clearly, our work and the study by Pinto and Pinto [18] show advantages of this line of thinking, rather than utilizing the traditional $EI_{mean}$ approach.

A few additional points and limitations are worthy of consideration. While our study did not seek to identify a single EI band that provides optimal data, Figs 3 and 4 revealed that most of the pixels within the EI histogram were between the range of 26–150 A.U. Like the work of Pinto and Pinto [18], our findings revealed that almost no useful muscle quality data could be

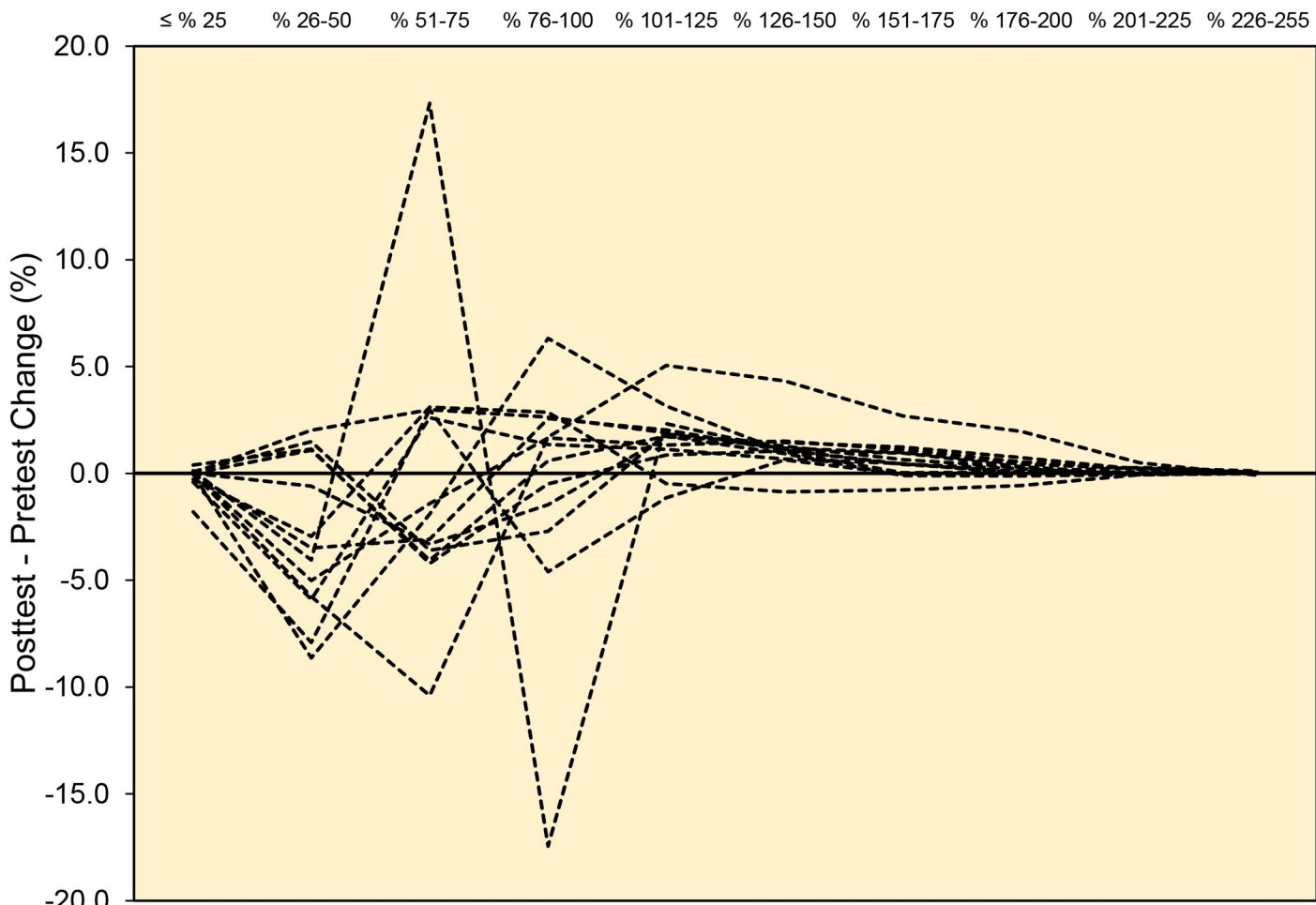

**Fig 4. Individual participant change score values across each of the 25 A.U. bands.**

obtained beyond 150 A.U. This finding seems to agree with a variety of studies conducted in older adults who, despite showing worse muscle quality than younger cohorts, typically show $EI_{mean}$ values $\leq$ 150 A.U. [26–28]. It should be noted, however, that the results of the present study are exclusive to the vastus lateralis, young females, the conditions provided by the study design, and the B-mode ultrasonography device and settings utilized by our laboratory. By comparison, the work by Pinto and Pinto [18] studied the rectus femoris muscle and utilized a different ultrasound device, rendering the methodological approaches of our studies difficult to compare. Finally, while the results of this study are intriguing from a methodological perspective, the exact anatomical and physiological underpinnings of changes in EI following various interventions are still unclear [29]. Caution should be taken in drawing bold conclusions about changes in specific EI bands being related to specific tissue adaptations.

## Conclusions

Two weeks of left knee joint immobilization in young females resulted in non-homogenous changes across distinct bands of the EI histogram for the vastus lateralis muscle. We observed

a large decrease in the percentage of pixels in the %EI$_{0-50}$ band, with significant differences across the change scores of other bands. These findings further the work by Pinto and Pinto [18] and suggest that analysis of distinct regions of the EI histogram, rather than simply analyzing the mean value, provides unique insights into changes in muscle quality during clinical interventions. Analysis of our data revealed that almost all the information within the EI histogram was between the range of 26–150 A.U, with 0–25 and 151+ A.U. providing little useful data concerning vastus lateralis muscle quality. Based on these findings, we encourage investigators to think critically about the robustness of data obtained from EI histograms, rather than simply reporting the mean value, in muscle quality research.

## Supporting information

**S1 Data.**
(XLSX)

## Author Contributions

**Conceptualization:** Zachary S. Logeson, Rob J. MacLennan, Gerard-Kyle B. Abad, Molly R. Gradl, Matheus D. Pinto, Ronei S. Pinto, Matt S. Stock.

**Data curation:** Rob J. MacLennan, Matt S. Stock.

**Formal analysis:** Zachary S. Logeson, Gerard-Kyle B. Abad, Johnathon M. Methven, Molly R. Gradl, Matt S. Stock.

**Funding acquisition:** Zachary S. Logeson, Rob J. MacLennan, Matt S. Stock.

**Investigation:** Rob J. MacLennan, Matt S. Stock.

**Methodology:** Rob J. MacLennan, Johnathon M. Methven, Molly R. Gradl, Matheus D. Pinto, Ronei S. Pinto, Matt S. Stock.

**Project administration:** Matt S. Stock.

**Resources:** Matt S. Stock.

**Supervision:** Matheus D. Pinto, Ronei S. Pinto, Matt S. Stock.

**Writing – original draft:** Zachary S. Logeson, Matt S. Stock.

**Writing – review & editing:** Zachary S. Logeson, Rob J. MacLennan, Gerard-Kyle B. Abad, Johnathon M. Methven, Molly R. Gradl, Matheus D. Pinto, Ronei S. Pinto, Matt S. Stock.

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
