## [Decision Letter · Decision Letter 0]

12 Nov 2021

PONE-D-21-29548The Impact of Skeletal Muscle Disuse on Distinct Echo Intensity Bands: A Retrospective AnalysisPLOS ONE

Dear Dr. Stock,

Thank you for submitting your manuscript to PLOS ONE. After careful consideration, we feel that it has merit but does not fully meet PLOS ONE’s publication criteria as it currently stands. Therefore, we invite you to submit a revised version of the manuscript that addresses the points raised during the review process. Please submit your revised manuscript by Dec 27 2021 11:59PM. If you will need more time than this to complete your revisions, please reply to this message or contact the journal office at plosone@plos.org. Please include the following items when submitting your revised manuscript:A rebuttal letter that responds to each point raised by the academic editor and reviewer(s). You should upload this letter as a separate file labeled 'Response to Reviewers'.A marked-up copy of your manuscript that highlights changes made to the original version. You should upload this as a separate file labeled 'Revised Manuscript with Track Changes'.An unmarked version of your revised paper without tracked changes. You should upload this as a separate file labeled 'Manuscript'.

We look forward to receiving your revised manuscript.

Kind regards,

Emiliano Cè

Academic Editor

PLOS ONE

Journal Requirements:

“Funding support for this project was provided to RJM by the De Luca Foundation and the University of Central Florida's Office of Research to MSS. Article processing charges were provided to ZSL in part by the UCF College of Graduate Studies Open Access Publishing Fund. The funders had no role in study design, data collection and analysis, decision to publish, or preparation of the manuscript.”

We note that you have provided funding information within the Acknowledgements Section. Please note that funding information should not appear in the Acknowledgments section or other areas of your manuscript. We will only publish funding information present in the Funding Statement section of the online submission form.

 “Funding support for this project was provided to RJM by the De Luca Foundation and the University of Central Florida's Office of Research to MSS. Article processing charges were provided to ZSL in part by the UCF College of Graduate Studies Open Access Publishing Fund. The funders had no role in study design, data collection and analysis, decision to publish, or preparation of the manuscript.”

Reviewers' comments:

Reviewer's Responses to Questions

**Comments to the Author**

1. Is the manuscript technically sound, and do the data support the conclusions?

Reviewer #1: Yes

Reviewer #2: Yes

2. Has the statistical analysis been performed appropriately and rigorously? 

Reviewer #1: Yes

Reviewer #2: Yes

3. Have the authors made all data underlying the findings in their manuscript fully available?

Reviewer #1: Yes

Reviewer #2: Yes

4. Is the manuscript presented in an intelligible fashion and written in standard English?

Reviewer #1: Yes

Reviewer #2: Yes

5. Review Comments to the Author

Reviewer #1: In this retrospective study, the Authors aimed at exploring whether the increase in ultrasound-derived echo intensity (EI) following two weeks of knee joint immobilization was limited to specific bands of the EI histogram.

Generally, this manuscript is well written and contributes to deepening our understanding of muscle quality by means of ultrasound technique and image analysis.

I have few more suggestions to improve the manuscript quality.

L18. I don’t think this sentence is completely correct. EI is measured from ultrasound. I suggest stating something like “… EI can be assessed quickly from ultrasound scans”.

L18-22. This is true if ImageJ is used as a software for the analysis. This should be acknowledged.

L103-106. Please, provide further details about the technique used for EFOV images acquisition.

L189. At this point of the manuscript there is no need to state that a difference is “significant”. I suggest erasing it.

L201. I am not a native English speaker, so consider this suggestion only if it is correct: “… the work by Pinto and Pinto..”

L209. Can you make a speculation about it?

L229. Same as at L201.

L245. Same as at L201.

Reviewer #2: The present manuscript aimed to investigate changes in muscle quality after two weeks of lower limb immobilization comparing the traditional EI mean with EI bands approach. The paper is well written and the methodological research question is of interest.

I only have some minor concerns that need to be addressed:

Abstract

I think your abstract conclusion should highlight also the comparison between 25 and 50 band intervals.

In the last sentence of the abstract, please specify that you are referring specifically to EI mean, otherwise this sentence could be misinterpreted (as EI is not robust in general)

L46-48: have you tried to carry out the correlation between VL CSA changes and EI in the different bands?

L209: Could you add at least some speculation on the potential physiological reasons?

L233: Why did you limit your analysis to VL? From Figure 2 the rectus femoris (RF) seems to be fully visible. I think that a comparison between VL and RF would be a nice addition to this work, considering that RF (being multi-joint) is generally less affected by disuse compared to the VL.

6. PLOS authors have the option to publish the peer review history of their article (what does this mean?). If published, this will include your full peer review and any attached files.

Reviewer #1: No

Reviewer #2: No

---

## [Author Response · Author response to Decision Letter 0]

23 Nov 2021

Please see our enclosed response document.

---

## [Decision Letter · Decision Letter 1]

29 Dec 2021

The Impact of Skeletal Muscle Disuse on Distinct Echo Intensity Bands: A Retrospective Analysis

PONE-D-21-29548R1

Dear Dr. Stock,

We’re pleased to inform you that your manuscript has been judged scientifically suitable for publication and will be formally accepted for publication once it meets all outstanding technical requirements.

Kind regards,

Emiliano Cè

Academic Editor

PLOS ONE

Additional Editor Comments (optional):

Reviewers' comments:

Reviewer's Responses to Questions

**Comments to the Author**

1. If the authors have adequately addressed your comments raised in a previous round of review and you feel that this manuscript is now acceptable for publication, you may indicate that here to bypass the “Comments to the Author” section, enter your conflict of interest statement in the “Confidential to Editor” section, and submit your "Accept" recommendation.

Reviewer #1: All comments have been addressed

2. Is the manuscript technically sound, and do the data support the conclusions?

Reviewer #1: Yes

3. Has the statistical analysis been performed appropriately and rigorously? 

Reviewer #1: Yes

4. Have the authors made all data underlying the findings in their manuscript fully available?

Reviewer #1: Yes

5. Is the manuscript presented in an intelligible fashion and written in standard English?

Reviewer #1: Yes

6. Review Comments to the Author

Reviewer #1: The Authors have successfully addressed all my comments and concerns. I do believe that the manuscript has improved.

7. PLOS authors have the option to publish the peer review history of their article (what does this mean?). If published, this will include your full peer review and any attached files.

Reviewer #1: No

---

## [Editor Report · Acceptance letter]

3 Jan 2022

PONE-D-21-29548R1 

The Impact of Skeletal Muscle Disuse on Distinct Echo Intensity Bands: A Retrospective Analysis 

Dear Dr. Stock:

I'm pleased to inform you that your manuscript has been deemed suitable for publication in PLOS ONE. Congratulations! Your manuscript is now with our production department. 

Kind regards, 

on behalf of

Professor Emiliano Cè 

Academic Editor

PLOS ONE